# Mitral Paravalvular Leak: Clinical Implications, Diagnosis and Management

**DOI:** 10.3390/jcm11051245

**Published:** 2022-02-25

**Authors:** Ignacio Cruz-Gonzalez, Pablo Antunez-Muiños, Sergio Lopez-Tejero, Pedro L. Sanchez

**Affiliations:** 1Department of Cardiology, University Hospital of Salamanca, 37007 Salamanca, Spain; cruzgonzalez.ignacio@gmail.com (I.C.-G.); ser_slt@hotmail.com (S.L.-T.); pedrolsanchez@me.com (P.L.S.); 2Instituto de Investigación Biomédica de Salamanca (IBSAL), 37007 Salamanca, Spain; 3Centro de Investigación Biomédica en Red Enfermedades Cardiovasculares (CIBERCV), 28029 Madrid, Spain

**Keywords:** paravalvular leaks, mitral regurgitation, heart failure and haemolytic anaemia, valvular prosthesis, percutaneous closure

## Abstract

Paravalvular leak incidence after mitral surgical replacement ranges from 7% to 17%. Between 1% and 5% of these are clinically significant. Large PVLs can cause important clinical manifestations such as heart failure or haemolysis. Current guidelines consider that surgical reparation is the gold-standard therapy in symptomatic patients with paravalvular leak. However, these recommendations are based in non-randomized observational registries. On the other hand, transcatheter paravalvular leak closure has shown excellent results with a low rate of complications, and nowadays it is considered the first option in selected patients in some experienced centres. In this review, we summarize the clinical manifestations, diagnosis, procedural details, and results of transcatheter mitral PVL closure.

## 1. Introduction

Paravalvular leak (PVL) is defined as the presence of any channel between the anatomical annulus and the prosthetic valve that causes a regurgitation jet between two chambers of the heart. As life expectancy continues to grow in developed countries, one of the consequences is that valvular heart disease is progressively more and more common. For example, in the United States or Europe, these pathologies affect up to 2.5% of the population, and prevalence is much higher in patients older than 75 years old. When severe valvular disease is established, surgical percutaneous valve replacement is usually needed. In Europe, mitral valve regurgitation and stenosis comprised 21% and 5% of all referrals for valve interventions. On the other hand, over recent years, in the United States more than 120,000 procedures, have been performed including at least 14,000 mitral surgical valve replacements [1]. After surgical intervention, different registries showed rates of PVL between 5% and 17%. Incidence of mitral valve replacement is higher than aortic ones, ranging from 7% to 17% and 2% to 10%, respectively [2,3]. The vast majority of these are diagnosed in the first year after surgery. Different risk factors of PVL have been identified such as heavy calcification of the annulus, the use of mechanical valves, non-pledged or continuous suture, endocarditis infection, larger atria or renal insufficiency. However, most PVLs are small and patients can remain asymptomatic. On the other hand, between 1 and 5% of them are clinically significant. Large PVLs can cause important clinical manifestations such as heart failure in almost 90% of the cases, or haemolysis in one-third of them, approximately.

European Society of Cardiology (ESC) and American Heart Association (AHA) guidelines on management of valvular heart disease still consider redo surgery as the first option for symptomatic patients with a PVL [4,5]. Nevertheless, they also emphasize that transcatheter closure should be considered depending on the surgical risk and local expertise. Furthermore, this new option has achieved, in different registries, excellent success outcomes that will be presented later in this review, with a very low rate of complications, in contrast with a second surgery.

## 2. Clinical Manifestations

As mentioned before, the majority of PVLs are trivial or mild, without causing any manifestation, so they remain underdiagnosed. However, some of them will be haemodynamically significant and will trigger different symptoms [3]. The diagnoses of these PVLs are usually performed within the first year after surgery, except for those in relation to endocarditis that could appear at any time.

The most common clinical manifestation is heart failure, especially in the largest ones. Mitral regurgitation will increase both intracavitary left chambers pressures, worsening the NYHA class function or developing pulmonary oedema. When it persists through the time, cardiac remodelling could appear and left atrium and ventricle volumes increase, pulmonary hypertension appears and even right-heart chambers may be affected.

Physical examination is very important in these cases. A new cardiac murmur can help us to suspect this complication, but it is not a very specific tool. Furthermore, during examination, we can find signs of heart failure such as rales or peripheral oedema.

Haemolysis is the second most frequent manifestation. In contrast with aortic leaks, haemolysis rate is higher in the mitral ones because they occur in systole, which implies a higher velocity of the regurgitation jet. Finally, anaemia may appear which increases the probability of new acute heart failure episodes.

Blood shear stress due to the PVL could also provoke more anaemia because of acquired Von Willebrand (VW) syndrome [6]. VW proteins are crucial to prevent bleeding because they act with platelets in thrombus formation. In this scenario, VW proteins lose their function, the coagulation process is altered and small blood losses may appear, increasing the risk of anaemia.

Blood tests can be helpful to suspect this problem. Not only will there be high levels of bilirubin, but also other analytical parameters will be altered. Low haptoglobin and high LDH (lactated dehydrogenase) will be observed because of erythrocytes’ destruction. It is possible to observe iron deficiency too, and reticulocytes and schistocytes rate will be increased [1]. Furthermore, cardiac markers of heart failure such as NT-proBNP increase and renal function may be altered. Products derived from haemoglobin are nephrotoxic and they can provoke renal failure, therefore the eGFR (estimated glomerular filtration rate) gets worse. Additionally, heart failure may affect it, as lower cardiac output or renal congestion can develop into a cardiorenal syndrome and will reduce renal filtration.

Identifying all these clinical and analytical manifestations and connecting them to PVL is vital in order to try to correct them and ameliorate patients´ symptoms.

## 3. Diagnosis

Diagnosis and characterization of PVLs are challenging. As we have discussed, it should be suspected when an onset of abnormal murmur appears at physical examination after a valvular replacement, especially in those patients who were admitted due to heart failure and/or haemolytic anaemia. In this situation, a transthoracic echocardiography (TTE) should be performed firstly. Rergardless, multimodal imaging is instrumental in guiding diagnostic and therapeutic strategies when managing PVL. These imaging techniques are summarized below and in the central illustration (Figure 1).

### 3.1. TTE

TTE is the initial diagnostic test of choice for all patients with suspected PVL. Although TTE is an excellent method for the assessment of valvular gradients, it is often limited by acoustic shadowing from mechanical components of prosthetic valves, annular calcification or prosthetic valve sewing rings [3]. Acoustic shadowing affects visualization of prosthetic valve components, and it may also result in the absence of colour Doppler signal with potential underestimation of the degree of PVL (Figure 2). This makes more difficult to identify the gradation of PVL [7]. At this point, Doppler evaluation can be a good tool to avoid underestimating PVL. In addition, a cardiac evaluation of atrial and ventricular size and function, pulmonary artery systolic pressure, and concomitant native valvular disease must be performed. It is important to investigate the presence of endocarditis due to its potential association with PVL.

### 3.2. TEE

Transoesophageal echocardiography is the gold standard when performing an exhaustive analysis of the PVL that can further characterize the leak regurgitation location, size, and severity [8]. Two-dimensional (2D) TEE is very sensitive in identifying the presence of PVLs; however, assessing the number, shape and location can be difficult in some cases [2]. Three-dimensional (3D) TEE achieves better definition, and it has been shown to be superior to 2D-TEE to study PVLs [9,10,11]. 3D images allow us to find out the shape (crescent-shaped vs. round), valve dehiscence, the distance from the sewing ring, the orientation and movement of prosthetic leaflets and the degree of regurgitation as well as helping us to improve the identification and quantification of multiple regurgitant jets [12]. Indeed, 3D-TEE is the recommended technique to guide percutaneous PVL closure procedures (especially in mitral location), as well as playing an important role in selection of the most appropriate closure device [8,11,13,14]. Recently, photorealistic rendering views (True- Vue, Philips Healthcare, Best, Netherlands) have been developed to increase 3D perception, making it possible to change the lighting source to improve contrast and enhance details, which would make it easier to identify defects [15,16] (Figure 3).

Nowadays, the clockwise format from “surgical view” is used to improve communication between interventional cardiologist and imaging specialists. In this scheme, the 12 o’clock position is at the mitral–aortic continuity, the left atrial appendage corresponds to the 9 o’clock position, and the interatrial septum is adjacent to the 3 o’clock position [17] (Figure 4).

The approach for detecting and grading prosthesis regurgitation is described in Table 1 and involves the evaluation of several echo parameters [3,17,18]:

Mitral PVL: qualitative parameters are used for mitral paravalvular regurgitation such as colour-flow regurgitant jet area, jet density, and systolic pulmonary venous flow reversal, a specific sign of severe mitral regurgitation. Due to the Coanda effect, the jet may be underestimated by jet area measurement. Quantitative parameters, such as vena contracta diameter and regurgitant volume and fraction, are also helpful. Although the proximal isovelocity surface area (PISA) approach has not been validated in the setting of paravalvular regurgitation, the presence of a large PISA could be consistent with more severe regurgitation.

Intracardiac echography (ICE) is less useful due to presence of acoustic shadowing, but it may be useful in certain instances (for example, patients which cannot be anesthetized, or who have oesophageal problems that forbid TEE, etc.). It can be performed without general anaesthesia, making procedures shorter and safer and further enabling the treatment of patients that may have been turned down for intervention. Ruparelia N. et al. reached acceptable procedural success rates (77.8%) with similar functional improvement and no related complications [19].

### 3.3. CMR

Cardiac MRI has a limited role within the diagnostic of PVL. Virtually all prosthetic valves (including mechanical valves) can be imaged by CMR [20]. Phase-contrast velocity mapping is performed in the short-axis plane just distal to the prosthetic valve, with subsequent quantitation of regurgitant volume and regurgitant fraction [21]. It can be useful in the presence of multiple and eccentric leaks or acoustic shadows that reduce the accuracy of echocardiography [3].

### 3.4. Cardiac CT

The most important contribution of cardiac CT is the capacity for anatomical characterization of PVL in patients with significantly limited echocardiographic images, which has an important role in pre-procedural planning [3]. Helical CT acquisition is performed in multiple phases with contrast injection protocols, and a reconstruction of theses phases is then processed. With adjustment of opacity and colour and applying cut-planes it is possible to visualize the PVL in great detail [18]. Despite these advantages, some evidence suggests that cardiac CT has no impact on PVL detection in comparison with 2D TEE [22].

In terms of fusion imaging, development of computed tomography (CT)–fluoroscopy fusion imaging has allowed CT imaging to provide a valuable tool for guidance during percutaneous PVL closure [23]. With this technique, single-phase CT data are reconstructed into 3D images. After that, they is co-registered with fluoroscopy and the relevant structures (such as the cardiac chambers, valves, sternotomy, etc) are overlaid onto the fluoroscopy screen. The CT data remain merged to fluoroscopy with rotation of the C-arm, providing real-time 3D anatomic information during the procedure. CT–fluoroscopy fusion can facilitate access, wire crossing, and device deployment during PVL closure [17]. This tool may be an important way to perform PVL closure in cases with special anatomical considerations, TEE ultrasound disturbance or X-ray translucent prothesis (Figure 5).

In selected complex cases, 3D printed models could be useful in the pre-procedure planning [24] (Figure 6).

## 4. Management

### 4.1. Treatment

Medical management has a limited role in patients with symptomatic PVLs, as it cannot resolve the underlying cause.

Both ACC/AHA 2020 and ESC 2021 Guidelines consider that surgical reparation is the gold-standard therapy in symptomatic PVL [4,5]. However, the level of evidence in ESC guidelines is only C (expert consensus). Moreover, AHA bases its decision in non-randomized observational registries. On the other hand, transcatheter closure is recommended in those symptomatic cases with a high or prohibitive operatory risk. Both guidelines point out that the percutaneous optionshould be considered depending not only on the risk status of the patient but also in the leak morphology and the local expertise.

Transcatheter closure of PVL has shown excellent results in different cases with a low rate of complications. This approach is contraindicated in the presence of active endocarditis, prosthesis instability or large PVL affecting more than 30% of the circumference [8].

### 4.2. Devices

For this purpose, different devices can be used. Most of them consist of a waist with two discs at each end [2]. The Occlutech PLD has two different shapes, one is square and another is rectangular [25]. On the other hand, the Amplatzer Vascular Plug (AVP) was once designed to close peripheral vessels, but nowadays is the most frequently used device for PVL closure. The AVPII is the most common in the United States (US) and the AVPIII in Europe [1,26], as the AVPIII has not been approved by the FDA yet. Duct, atrial septal and muscular ventricular septal occluders can be also used for this procedure in selected cases. In this sense, the selection of the device must take in to account the shape and size of the PVL and the operators experience. AVPIII or Occlutech PLD devices may be used in oval leaks due to its rectangular shape. In contrast, square devices such as AVPII or Occlutech PLD can be useful in leaks with a more cylindrical shape. Those with a crescent shape could be closed with a rectangular device. These different devices are shown in Figure 7. In terms of sizing, we recommend to use devices at least 1–2 mm larger than the PVL maximum diameter; in some large PVLs, two devices can be deployed simultaneously, as previously described [8].

### 4.3. Procedure

Pre-procedural planning is one of the most important steps of a successful procedure. Localization and characterization of the leak are crucial, and the closure strategy must be decided before initiating the procedure based on the previous information. In our opinion, transcatheter closure of mitral PVL should be guided by 3D transoesophageal echocardiography under general anaesthesia

#### 4.3.1. Anterograde Approach

In the antegrade approach, once a successful transeptal puncture is performed, a diagnostic catheter (e.g., Judkins right) is advanced into the left atrium. In most cases, the use of a deflectable catheter (e.g., Agilis, Abbott medical) is recommended [3,8]. After that, a hydrophilic guidewire (e. g., Terumo guidewire, Terumo Medical Corporation) is often used to cross the mitral PVL, and in most cases, an arteriovenous loop is established in the aorta. Alternatively, an extra-support wire can be placed in the left ventricle. Finally, a delivery sheath is advanced from the venous access over the loop or extra-support wire and the device is deployed. Before releasing, the disc movement in mechanical valves should be confirmed (Figure 8).

#### 4.3.2. Retrograde Approach

In the retrograde approach, a hydrophilic guidewire (e.g., Terumo guidewire, Terumo Medical Corporation) over a catheter (e.g., Judkins right) is often used to cross the PVL from the left ventricle to the left atrium. After crossing, an arteriovenous wire loop is often created in the left atrium; therefore, a transeptal puncture is needed. Finally, the delivery sheath is advanced from the venous access and the device is deployed (Figure 9).

#### 4.3.3. Transapical Approach

Transapical access could be an alternative for mitral PVL closure (especially for posterior or septal defects or patients with mitro-aortic monodisc mechanical valves) [27] (Figure 10). The main advantages of this access are the often less difficult wiring of the PVL and less resistance to cross the PVL, however the rate of complications of this approach is higher than in the retrograde or antegrade approach.

### 4.4. Final Result Assesment

Before releasing the device, echocardiography reassessment should be made in order to discard complications, such as disc movement restriction or LVOT (left ventricle outflow tract) obstruction. In anterior and septal PVL, an extremely rare but severe complication could be LVOT obstruction. It should be suspected in patients with septal hypertrophy, and it must be assessed by echocardiography during the procedure. In case this complication occurs, the device should not be released and its orientation or size should be changed (i.e., two smaller devices can be deployed rather than one larger device). In the same way, disc movement blockage should be always checked before device releasing; this is particularly important in monodisc prosthesis. This complication could be prevented using smaller devices, changing the orientation of the device, deploying the “ventricular” disc inside the PVL tunnel or using devices such as the AVPII (the disc size is the same as the body of the device). However, these complications are very rare, but disc blockage is usually the reason for urgent surgery after PVL closure [28].

Furthermore, sometimes a residual leak is detected before releasing. In this situation, a simultaneous deployment of two devices can be performed, or a second device can be deployed sequentially [8]. If the leak is detected after releasing the device or during follow-up, this residual leak can be recrossed, and a second device can be deployed.

### 4.5. Post-Procedural Medical Therapy and Follow-Up

For patients under anticoagulation therapy, this should be continued after the procedure. Dual antiplatelet therapy for at least 3 months is recommended in non-anticoagulated patients (i.e., biological prostheses). Post-procedural imaging with TEE to assess device position and residual regurgitation is recommended. The timing of a follow-up TEE varies between institutions, but we recommend an initial early TEE 3 months after the procedure. If persisting or new PVL are observed, percutaneous or surgical management can be chosen depending on the size of the remaining PVL.

## 5. Outcomes

Redo surgery for PVL closure has been demonstrated to improve outcomes. However, it offers high morbidity and mortality rates. Redo surgery has elevated rate of complication and leak closure success decreases with any extra intervention [8]. On the other hand, the percutaneous method for PVL closure has emerged as a safer intervention maintaining those good results [28]. Not only is technical success rate high, between 77% and 91%, but also clinical benefit ranges from 66% to 77% depending of the registries [8]. Leak closure seems to ameliorate NYHA class from 2.7 ± 0.8 to 1.6 ± 0.8 after PVL closure in only a median of 110 days of follow-up (*p* < 0.001), as reflected in the UK and Irish registry [29]. Moreover, significant haemolysis rate after the procedure is very low, around 1.6%. A metanalysis directed by Millan et al. showed that in addition to a better functional class, successful PVL closure was demonstrated to reduce cardiovascular mortality (OR = 0.08, CI95% 0.01–0.90), with a positive trend in the overall mortality, and it also reduces haemolytic anaemia (OR = 9.95, CI95% 2.10–66.73) [30]. Furthermore, complication rates are much lower than in redo surgery. The most frequent ones are device embolization, stroke, cardiac perforation or vascular complications at the access site [2]. Table 2 summarizes studies that compare percutaneous versus surgical closure. Even more, these results will keep improving in the future as operators’ experience continues to grow and in selected centres transcatheter closure becomes considered as the first-line therapy for selected cases.

## 6. Conclusions

PVL is a frequent complication of surgical valve replacement, with significant morbidity and mortality. Emerging data indicate that percutaneous closure of PVL is a safe and effective alternative to surgical closure. This procedure has become the first-line treatment in clinical practice in experienced centres.

## Figures and Tables

**Figure 1 jcm-11-01245-f001:**
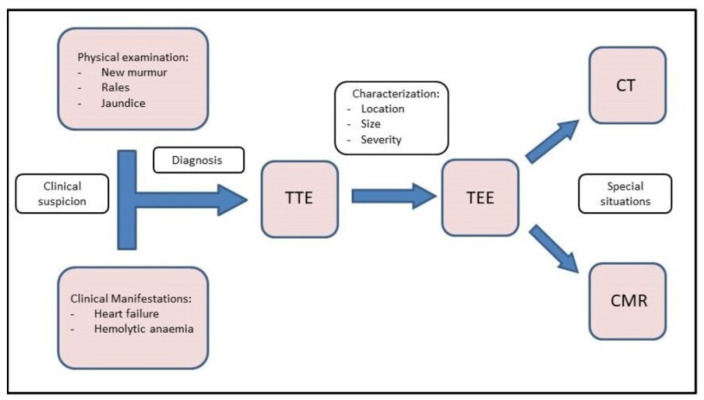
Central Illustration. Diagnostic flow chart. Diagnosis starts with clinical suspicion. TTE and TEE should be performed to confirm the diagnosis and correct characterization. Finally, CMR and CT can be used in special situations.

**Figure 2 jcm-11-01245-f002:**
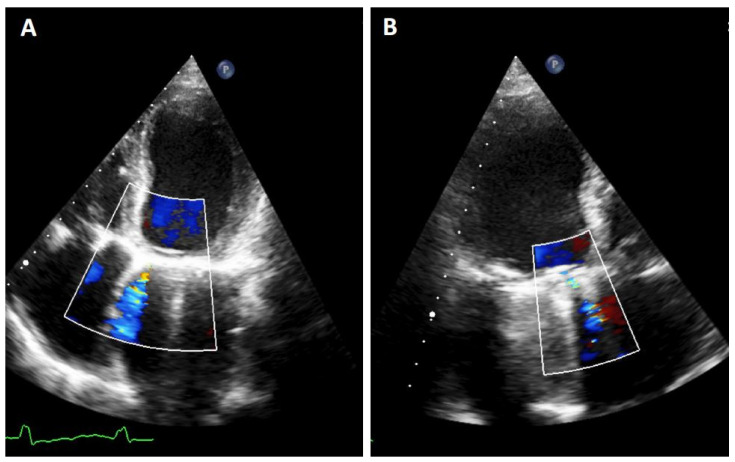
TTE PVL. Mitral regurgitation is detected. It must be noted that it is difficult to quantify the exact proportion of the flow in both planes due to artifacts as acoustic shadow. (**A**): 4-chamber image showing a mitral PVL with colour Doppler. (**B**): 3-chamber with acustic shadoiw in the left atrium and an anterior regurgigant jet of a PVL leak.

**Figure 3 jcm-11-01245-f003:**
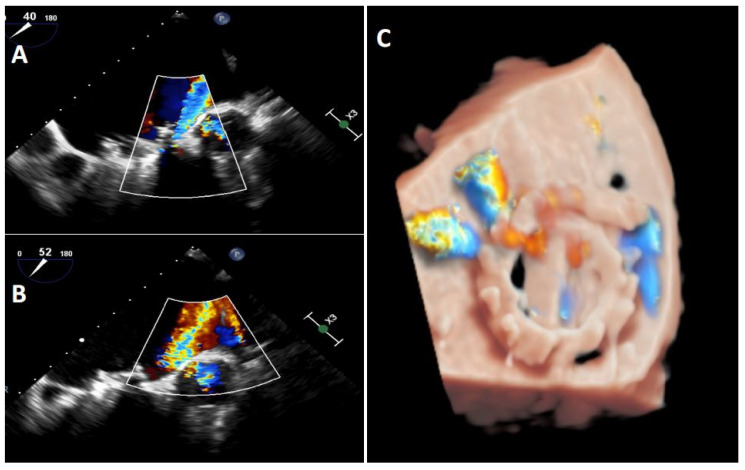
Mitral TEE. (**A**,**B**): regurgitation jet more severe that we could see at TTE on figure (**A**); (**C**): Truevue with colour Doppler.

**Figure 4 jcm-11-01245-f004:**
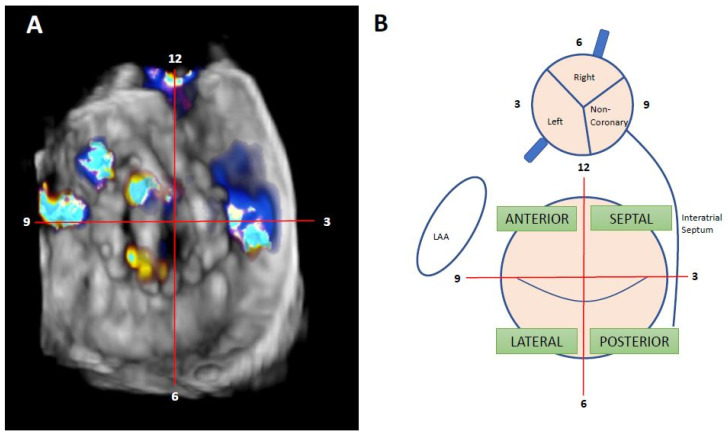
Localization of mitral PVL. (**A**): Clockwise format. Surgeon’s view. TEE from the same patients; we can see the presence of two anterolateral leaks, at 9 and 10 h, and one septal leak at 3 h. (**B**): mitral and aortic drawing of clockwise format and interactions between different hearts structures—adapted from reference [2].

**Figure 5 jcm-11-01245-f005:**
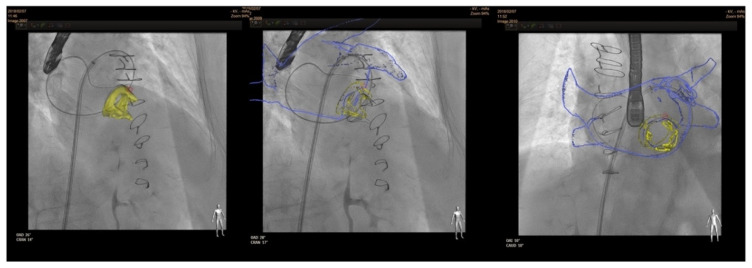
CT–fluoroscopy fusion for mitral PVL closure.

**Figure 6 jcm-11-01245-f006:**
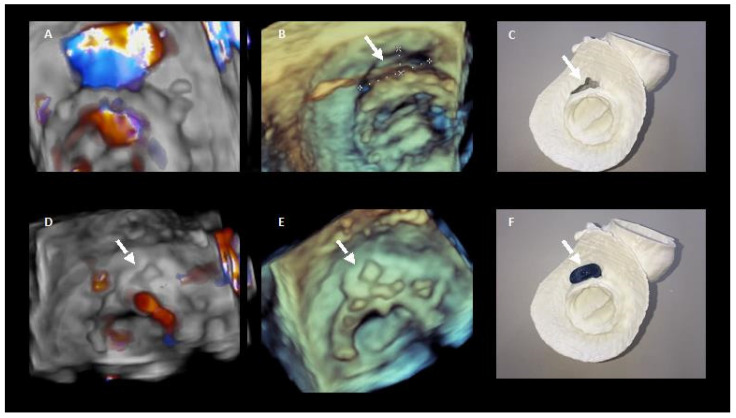
3D printing in preprocedural planning of paravalvular leak closure. (**A**,**B**). 3D echocardiography images of the PVL. (**D**,**E**) 3D echocardiography images showing the result after percutaneous closure. (**C**,**F**) show the 3D model of the PVL and how it colud be closed with the device.

**Figure 7 jcm-11-01245-f007:**
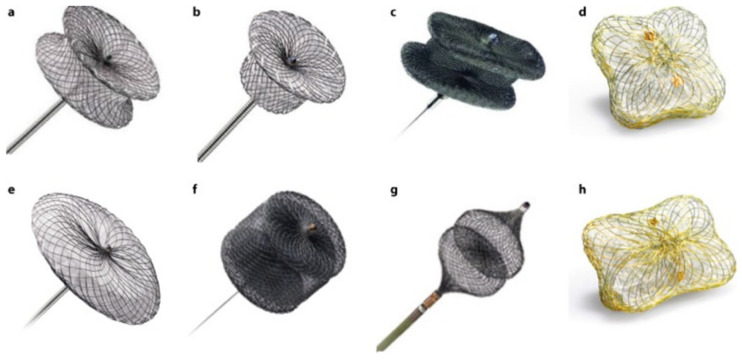
PVL closure devices. (**a**): Amplatzer Muscular VSD Occluder. (**b**): Amplatzer Duct Occluder. (**c**): Amplatzer Vascular Plug III. (**d**): Occlutech PLD (square-shaped design). (**e**): Amplatzer Septal Occluder. (**f**): Amplatzer Vascular Plug II. (**g**): Amplatzer Vascular Plug IV. (**h**): Occlutech PLD (rectangular-shaped design).

**Figure 8 jcm-11-01245-f008:**
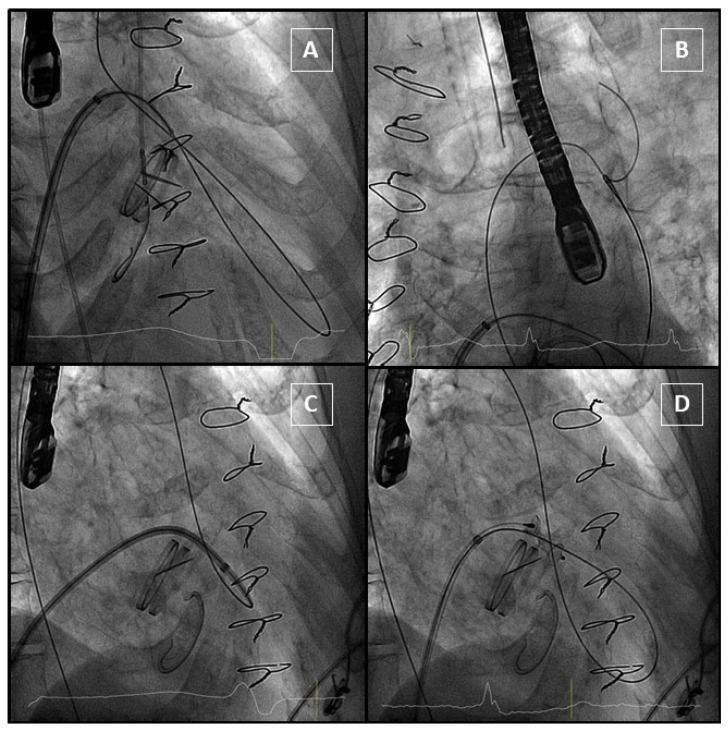
Anterograde approach. (**A**): Hydrophilic guidewire passes through the PVL and gets into aorta. (**B**): Guidewire is snared from an arterial access to complete the arteriovenous loop. (**C**): The delivery sheath crosses the PVL. (**D**): The device is delivered, and it does not interfere with the mechanical prosthesis.

**Figure 9 jcm-11-01245-f009:**
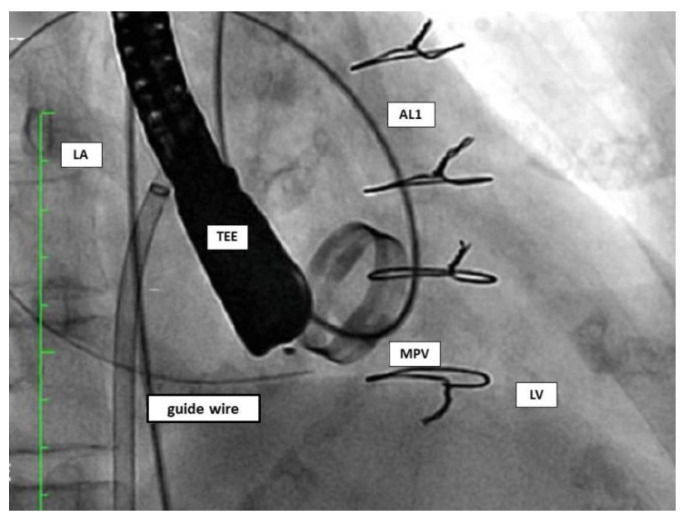
Retrograde approach. AL1: Amplatz Left catheter; LA: left atrium; LV: left ventricle; MPV: mitral prosthetic valve; TEE: transoesophageal echocardiography.

**Figure 10 jcm-11-01245-f010:**
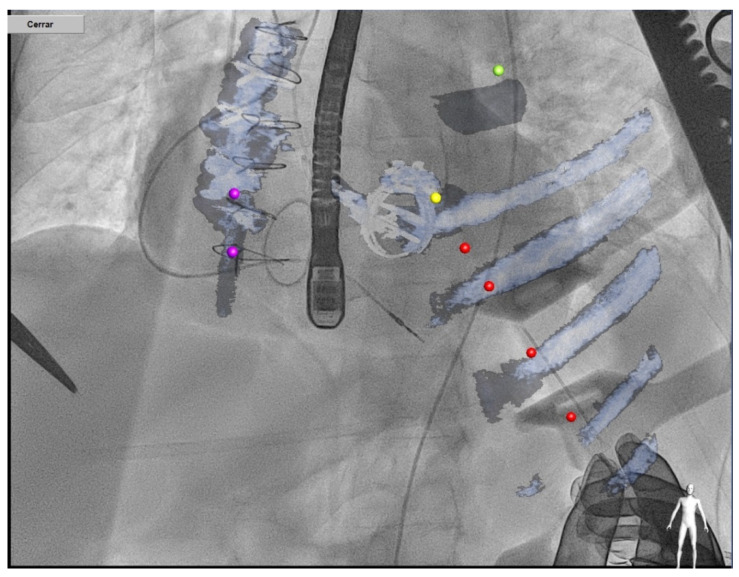
Transapical approach using CT–fluoroscopy fusion.

**Table 1 jcm-11-01245-t001:** Assessment of PVL severity.

	MILD	MODERATE	SEVERE
Colour Flow Area	<4 cm^2^, <20% LA area	Variable	>8 cm^2^, >40% LA area
Jet Density	Incomplete	Dense	Dense
Jet Contour	Parabolic	Variable	Early peaking, triangular, holosystolic
Pulmonary Venous Flow	Normal	Systolic blunting	Systolic flow reversal
PASP	Normal	Variable	Incremented
Vena contracta	<3 mm	3–6.9 mm	>7 mm
Circumferential extent of PVL	<10%	10–29%	>30%
Regurgitant Volume	<30 mL	30–59 mL	>60 mL
Regurgitant Fraction *	<30%	30–49%	>50%
EROA	<20 mm^2^	20–39 mm^2^	>40 mm^2^

Mitral PVL quantification criteria. * Cardiac MR has the same values for this parameter. Adapted from reference [3].

**Table 2 jcm-11-01245-t002:** Percutaneous vs. surgical PVL mitral closure.

Study	Country and Period	Type of Study	N Percutaneous vs. Surgical Closure	Endpoint	Results
Tamarasso et al., 2014	Italy 2000–2013	Single-centre, retrospective analysis	17 vs. 122	In-hospital death	Risk of death increased with surgical treatment (OR 8.0, 95% CI 1.8–13; *p* = 0.05)
Angulo-Llanos et al., 2016	Spain 2008–2014	Single-centre, retrospective, propensity-score matched analysis	51 vs. 36	Composite of death or readmission. (mean follow-up 784 days)	- Non-significant difference in composite end point. - Reduced in-hospital mortality with percutaneous approach.
Pinheiro et al., 2016	Brazil 2011–2013	Single-centre, retrospective analysis	10 vs. 25	Reintervention or death at 1 year	Non-significant difference between groups for either end point
Milan et al., 2017	Canada 1994–2014	Single-centre, retrospective, propensity-score matched analysis	80 vs. 151	Composite of all-cause death and hospitalization for heart failure. Median follow-up 3.5 years	Reduced risk of end point with surgical treatment (HR 0.28; 95% CI 0.18–0.44; *p* < 0.001)
Alkhouli et al., 2017	USA 1995–2015	Single-centre, retrospective analysis	195 vs. 186	Technical success and long-term survival (mean follow-up 4 years)	- Technical success greater in the surgical group - Non-significant difference in long-term survival between groups.
Wells et al., 2017	USA 2007–2016	Single-centre, retrospective analysis	56 vs. 58	Composite of death, reintervention or heart failure admission at 1 year	No difference in primary end point or 1-year survival between groups
Zhang et al., 2017	China 2009–2015	Single-centre, retrospective analysis	46 vs. 41	Survival Mean follow-up 49 months	- Non-significant difference in survival - Fewer in-hospital major adverse events and more cost-effective with percutaneous treatment

Summarize of studies comparing percutaneous vs. surgical closure of mitral PVL.

## Data Availability

Not applicable.

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
