# Peer review of "Mitral Paravalvular Leak: Clinical Implications, Diagnosis and Management"

_jcm, 2022, doi:10.3390/jcm11051245_

Round 1

Reviewer 1 Report

I read with interest the manuscript number jcm-1571621 entitled “Mitral paravalvular leak: clinical implications, diagnosis and management” by Cruz-Gonzalez and colleagues.

This review summarizes the clinical manifestations, diagnosis, procedural details, and results of trans-catheter mitral paravalvular leak closure (PVL).

The manuscript focuses on an interesting and up-to-date topic.

I have the following comments:

  • The epidemiological data provided in the “Introduction” section mainly refer to the US experience which is sub-optimal and unbalanced given the amount of European data available to date, also taking into account that the manuscript comes from a European group. Please provide a snapshot including a wider spectrum of data.
  • A central illustration with a schematic diagnostic flow-chart might have improved the readability of the manuscript.
  • The “Devices” 4.2 section is mostly felt as a bare list by this reviewer; a more detailed description of such devices, focusing on the most common morphological and procedural aspects leading to the most appropriate device selection (shapes and sizes), would improve the quality and usefulness of the review.

Author Response

  • The epidemiological data provided in the “Introduction” section mainly refer to the US experience which is sub-optimal and unbalanced given the amount of European data available to date, also taking into account that the manuscript comes from a European group. Please provide a snapshot including a wider spectrum of data.

Response: Epidemiological data about the prevalence of valvulopathies available in the literature are similar in all industrialized countries, including the USA and European countries. A new snapshot including European data has been added. Moreover, data provided about PVL are already reflecting both American and European data.

  • A central illustration with a schematic diagnostic flowchart might have improved the readability of the manuscript.

Response: A central illustration has been added as you suggest.

  • The “Devices” 4.2 section is mostly felt like a bare list by this reviewer; a more detailed description of such devices, focusing on the most common morphological and procedural aspects leading to the most appropriate device selection (shapes and sizes), would improve the quality and usefulness of the review.

Response: There are now explained in the Devices section, some morphological aspects, and recommendations.

Reviewer 2 Report

This manuscript is about the intervention management of ‘mitral paravalvular leakage’ Currently, vascular plug occlusion for mitral paravalvular leakage is a trend. However, the recommendation and level of evidence for intervention treatment is not clear. So I suggest some comments and questions to authors.

  1. For anterior or setal side leak, vascular plug might can be an obstacle for LVOT flow because of opposite side positioning over aortomitral curtain. So we have to select carefully patients for anatomically acceptable cases for prevention flow complication. So I suggest for authors to clearly describe a selection criterion for interventional treatment for mitral paravalvular leakage in institution.
  2. Sometimes, after intervention treatment, vascular plug can interrupt the mechanical mitral valve disc movement, so, hemodynamically, plug can make a mitral stenosis. So how many cases did authors experience that kind complications after interventions and have to describe some possible complications.
  3. In some cases, paravalvular leak range is larger and longer than predicted size and location from imaging studies such as CT scan, echocardiogram. So after treatment, plug can make more detachment of artificial mitral valve sewing ring from annulus. So authors also have to describe possible complication and distinction tip for intervention case and surgical treatment case

Author Response

  • For anterior or setal side leaks, a vascular plug might be an obstacle for LVOT flow because of opposite side positioning over the aortomitral curtain. So we have to select carefully patients for anatomically acceptable cases for prevention flow complications. So I suggest for authors to clearly describe a selection criterion for interventional treatment for mitral paravalvular leakage in institutions.

Response: This complication is now remarked in the Procedure section to explain it as you suggest

  • Sometimes, after intervention treatment, vascular plug can interrupt the mechanical mitral valve disc movement, so, hemodynamically, plug can make mitral stenosis. So how many cases did authors experience that kind of complication after interventions and have to describe some possible complications.

Response: This complication is seen sometimes because of the interference of the device with the disc, blocking it, and not letting it close. In these cases, mitral regurgitation, stenosis, or both can be seen. The rate of this complication is unknown because it is easily noticed during the procedure and then the device is recaptured at that moment to avoid it. Valve disc movement is not normally detected after the procedure if there was a correct evaluation during it, as we emphasize in the text.

  • In some cases, the paravalvular leak range is larger and longer than the predicted size and location from imaging studies such as CT scans, echocardiograms. So after treatment, the plug can make more detachment of artificial mitral valve sewing ring from the annulus. So authors also have to describe possible complications and distinction tips for intervention case and surgical treatment case

Response: As remarked now in the article, in some cases only one device is not enough, so a second device must be needed (if a residual leak is detected at the end of the procedure). We have included in the manuscript this comment and we have detailed some techniques to manage this situation.

Round 2

Reviewer 1 Report

The manuscript is now more balanced and useful to the readers.